# Advanced Nano-Drug Delivery Systems in the Treatment of Ischemic Stroke

**DOI:** 10.3390/molecules29081848

**Published:** 2024-04-18

**Authors:** Jiajie Zhang, Zhong Chen, Qi Chen

**Affiliations:** 1Key Laboratory of Neuropharmacology and Translational Medicine of Zhejiang Province, School of Pharmaceutical Sciences, Zhejiang Chinese Medical University, Hangzhou 310053, China; 202321126811230@zcmu.edu.cn (J.Z.); chenzhong@zju.edu.cn (Z.C.); 2Interdisciplinary Institute for Medical Engineering, Fuzhou University, Fuzhou 350108, China

**Keywords:** nanoparticles, ischemic stroke, nano-drug delivery system, blood–brain barrier

## Abstract

In recent years, the frequency of strokes has been on the rise year by year and has become the second leading cause of death around the world, which is characterized by a high mortality rate, high recurrence rate, and high disability rate. Ischemic strokes account for a large percentage of strokes. A reperfusion injury in ischemic strokes is a complex cascade of oxidative stress, neuroinflammation, immune infiltration, and mitochondrial damage. Conventional treatments are ineffective, and the presence of the blood–brain barrier (BBB) leads to inefficient drug delivery utilization, so researchers are turning their attention to nano-drug delivery systems. Functionalized nano-drug delivery systems have been widely studied and applied to the study of cerebral ischemic diseases due to their favorable biocompatibility, high efficiency, strong specificity, and specific targeting ability. In this paper, we briefly describe the pathological process of reperfusion injuries in strokes and focus on the therapeutic research progress of nano-drug delivery systems in ischemic strokes, aiming to provide certain references to understand the progress of research on nano-drug delivery systems (NDDSs).

## 1. Introduction

With the improvement of living standards, people’s diets have become more refined, often high in oil, fat, and sugar. This has increased the burden on cardiovascular health, leading to a rise in conditions such as hypertension, diabetes, and atherosclerosis [1,2,3,4]. These diseases are major contributors to the increasing frequency of strokes year by year. Strokes have consequently become the second leading cause of death worldwide, attributed to its narrow therapeutic window, high mortality and disability rates, and high recurrence rate [5]. Strokes are divided into ischemic and hemorrhagic strokes, with ischemic strokes (ISs) accounting for 71% of all strokes [6]. ISs are caused by vascular embolism, which leads to ischemia and hypoxia in brain tissue. The only FDA-approved treatment for an IS is tissue plasminogen activator (tPA), which is used to treat vascular embolisms by restoring blood flow through intravenous thrombolysis. However, a reperfusion injury has greater damage to cerebral neurons, triggering a cascade of events, mainly including, energy failure, loss of cellular ionic homeostasis, excitotoxicity, mitochondrial function impairment, reactive oxygen species (ROS) generation, and immune cell infiltration, resulting in neural inflammation [7,8,9].

As the world’s most populous country, China accounts for approximately one-third of global stroke deaths, posing a significant obstacle to people’s lives and health. Consequently, research on stroke treatment has become a prominent focus in China [10]. Currently, the main clinical modalities used for stroke treatment include surgical thrombolysis and pharmacological thrombolysis [11]. The intravenous thrombolysis of tPA within 4.5 h is effective in clinical treatment, and a thrombectomy performed within 24 h also yields benefits. However, these interventions carry a narrow therapeutic window and entail a certain risk of hemorrhage. Effective recovery is achieved in only a small percentage of patients due to time constraints. The reopening of blood flow can result in a reperfusion injury, substantial ROS production, and an inflammatory response that damages neurons [12,13,14]. To date, many studies report that the inhibition of neuroinflammation is effective in alleviating secondary injury after IS reperfusion [15,16]. The infiltration of immune cells, such as microglia, astrocytes, and NK cells, is the main cause of inflammation resulting from a reperfusion injury in ISs [17,18]. Among them, microglia mainly consist of M1 and M2 types, and it has been indicated that the transition from pro-inflammatory M1 microglia to anti-inflammatory M2 type is conducive to the recovery of stroke prognosis [19]. Moreover, the suppression of neuroinflammation and scavenging of ROS are key factors in enhancing prognostic recovery from strokes. Simultaneously, the salvage of ischemic penumbra neurons can be accomplished through the use of neuroprotective agents [20]. Edaravone, a free-radical scavenger commonly used in clinical treatment, can reduce BBB damage and inhibit inflammation. However, it faces significant challenges, including difficulty in crossing the BBB, poor bioavailability, and rapid clearance from the blood, all of which contribute to low drug efficacy [21]. To address these challenges, researchers have initiated studies in nanotechnology, aiming to convert drugs into nanoforms to enhance drug utilization and reduce side effects, thereby achieving the therapeutic goal [22].

For the past few decades, nanotechnology has been developing rapidly, and nano-drug delivery systems have shown great advantages [23]. Nano-drug delivery systems can improve the solubility of insoluble drugs, enhance the stabilization, and prolong the half-life in the body. The precise targeting and control of drug release could also be realized through additional targeting fragments or surface modifications in nano-drug delivery systems, ultimately achieving the purpose of increasing efficiency and reducing toxicity [24,25,26,27,28]. The BBB after an IS is somewhat compromised, with an increase in permeability, but not enough to fail completely and still have a strong ability to insulate against external substances. Fortunately, the application of nanotechnology shows large potential and a bright future for nano-drug delivery, facilitating improved drug utilization across the BBB and into the brain [29,30]. Interestingly, carriers can also treat diseases, for example, Prussian blue (PB), which has an ROS-scavenging effect [31]. This review briefly introduces the pathologic process of reperfusion in ISs and highlights recent studies of NDDSs in treating brain ischemic reperfusion injuries, aiming to offer references that could help bridge the gap between laboratory research and clinical application.

## 2. The Pathophysiology of ISs

Multiple complex pathological mechanisms are involved in brain damage caused by ISs [32,33,34]. Understanding these complex mechanisms not only deepens our understanding of the disorders but also aids researchers in facilitating the translation from the laboratory to clinical practice. At the onset of an IS, local ischemia and brain hypoxia lead to metabolic disorders in the intracerebral microenvironment, which then trigger a series of cascading injury processes, as shown in Figure 1 [35]. IS pathology begins with cerebral ischemia and hypoxia, culminating in neuronal apoptosis [36]. Energy failure is due to ischemia and hypoxia, which stimulate neuronal depolarization to release large amounts of glutamate. This process causes calcium influx and results in mitochondrial dysfunction [37]. At the same time, the accumulation of glutamate induces cellular neuro-excitotoxicity, the massive release of reactive oxygen and nitrogen species (RONS), and the penetration of peripheral immune cells to the brain [17,38,39]. After vascular reperfusion, oxygen and energy supply are restored, but mitochondrial dysfunction and the inability to process excess oxygen disrupt the balance between RONS production and clearance [40,41]. Excessive RONS would induce oxidative stress, which damages endothelial cells, and causes platelet aggregation and adhesion in microvessels, leading to the reformation of the thrombus and ultimately resulting in cerebral edema and bleeding [42,43,44]. This in turn further promotes immune cell infiltration, exacerbating the inflammatory response and causing neuroinflammation [45]. These reactions interact with each other and do not exist singularly. When an IS occurs, thrombolysis within a few hours can alleviate brain damage; however, secondary damage from cerebral ischemia–reperfusion poses a greater challenge [46,47].

Although cerebral neuronal damage in the core area of ischemia is irreversible, the penumbral area of cerebral ischemia affected by the spread of ROS and immune cell infiltration is recoverable [48,49]. Therefore, the semi-dark band in cerebral ischemic is an important target area for improving prognosis and restoring neurological function [5,50,51]. The cerebral ischemic penumbra can be rescued by suppressing the immune–inflammatory response and attenuating oxidative stress [52,53,54]. Some studies have found that administering exogenous mitochondria to the ischemic area can alleviate mitochondrial dysfunction. Exogenous mitochondria can use endogenous mechanisms to repair cellular damage, potentially resulting in therapeutic effects on the nerves after an IS [55,56,57]. 

In this section, we mainly focus on excitotoxicity, oxidative stress, and neural inflammation. Other mechanisms of injury will not be discussed further.

### 2.1. Excitotoxicity

Excitatory amino acids, such as glutamate and aspartate, serving as central neurotransmitters, play critical roles in transmitting messages in the nervous system [58,59]. Among them, glutamate is widely distributed throughout the brain and exerts a significant excitatory effect on central neurons [37]. Under physiological conditions, the brain microenvironment maintains homeostasis without excessive glutamate production, which can be metabolically regulated. However, when an ischemic stroke occurs, because the blood supply is interrupted, neurons are deprived of energy and oxygen, resulting in decreased ATP production. This leads to the abnormal functioning of the NA^+^-K^+^ ion pump in the cell membrane [60]. Meanwhile, N-methyl-D-aspartate receptors (NMDAR) regulate ion-gated pathways by binding glutamate, thereby exerting excitatory effects on neuronal cells [42,61]. Unfortunately, when substantial amounts of glutamate binding to aspartate receptors, it triggers a significant inward flow of Ca^2+^, resulting in calcium overload and NO^−^ production [62]. NO^−^ reacts with O^2−^ to produce the destructive OONO^−^, exacerbating BBB damage and brain injury [63,64,65]. In conclusion, excitotoxicity arises from uncontrolled glutamate release and calcium overload, triggering a series of effects that ultimately result in the death of neurons [66,67]. Thus, excitotoxicity represents an important molecular mechanism contributing to IS damage [68].

### 2.2. Oxidative Stress

As a downstream consequence of an excitatory injury, oxidative stress emerges as another major molecular mechanism contributing to ISs [23]. Oxidative stress occurs when the oxidative homeostasis system is disrupted due to the massive release of free radicals [69]. Excessive free radicals cause cellular damage, DNA damage, skeleton disruption, and lipid peroxidation, ultimately resulting in neuronal damage and brain death [70,71]. In particular, the free radical blowout after reperfusion exposes neurons to even more severe challenges [72,73]. In addition, free radicals can stimulate the secretion of cytokines as well as the expression of adhesion molecules, which mediate inflammatory and immune responses, thereby exacerbating a brain tissue reperfusion injury [74].

### 2.3. Neural Inflammation

The inflammatory response plays a pivotal role in the pathological events associated with brain damage and repair processes in ischemic strokes [75]. Various inflammatory cells participate in the inflammatory reaction. Microglia, astrocytes, neutrophils, NK cells, and other immune cells constitute the major effector cells during an IS [76,77]. Here, we take microglia and NK cells as examples. Microglia are intrinsic immune cells residing in the brain that play an important role in CNS diseases and that can be classified into M1 and M2 phenotypes [78,79]. M1 microglia secrete massive ROS, pro-inflammatory factors, and protein hydrolases to promote inflammation and exacerbate neuronal injuries, whereas M2 microglia secrete anti-inflammatory factors, which attenuate inflammatory injuries and improve the recovery of brain injuries [75,80]. While the activation of microglia is initially intended to protect neuronal cells, their overactivation can lead to harmful inflammation and neuronal death [81]. Numerous studies have shown that facilitating the conversion of microglia from M1 to M2 types can reduce inflammation and improve stroke prognosis [52,82,83,84,85]. For NK cells, they exacerbate stroke injuries by increasing local inflammation and neuronal hyperactivation, ultimately resulting in neuronal cell death [86,87]. The reason for this is that NK cells, which are common immune cells in the body, are actively recruited into the site of injury when an IS occurs. Simultaneously, NK cells secrete a large number of cytokines that promote inflammatory damage, ultimately leading to irreversible brain damage. Although immune cells that initially infiltrate into brain tissue exacerbate brain damage and neurological dysfunction, they also later play beneficial roles, such as promoting glial scarring and the phagocytosis of debris, which are essential for wound healing [88].

## 3. Advanced NDDSs for ISs

Prolonging the time window of treatment and mitigating a secondary reperfusion injury are ideal for addressing ISs [89]. Fortunately, recent advances in NDDSs have emerged, offering promising possibilities for enhancing IS reperfusion therapy [90]. NDDSs are known for their small size, large surface area, controlled release, targeted modification, and high ability to penetrate the BBB [91,92]. In addition, NDDSs can be utilized to load hydrophobic, hydrophilic, and gene-based drugs while also functioning as labeling probes for tracer imaging of ISs [93,94]. In the realm of advanced NDDSs for an ischemic stroke (IS), common types include polymers, inorganic nanoparticles, liposomes, and cell membrane-coated nanoparticles [95,96].

### 3.1. Polymers

Polymers offer broad prospects for addressing drug delivery for ISs [29]. They excel as NDDSs for treating CNS disorders due to their exceptional qualities, including excellent biodegradability, high biocompatibility, and minimal toxicity [97,98,99]. There are numerous polymers, among which PLGA, poly(ethylene glycol) (PEG), and poly(lactic acid) (PLA) are common and the most widely studied [100]. The appropriate modification of polymeric nanoparticles can enhance brain targeting and achieve drug enrichment in the target region [100]. PLGA is synthesized from the polymerization of two monomers: lactic acid and hydroxyacetic acid. It is an FDA-approved drug excipient known for its high encapsulation efficiency and excellent biocompatibility [101,102]. Previous studies by Wang et al. have demonstrated that epidermal growth factor (EGF) and erythropoietin (EPO) have a stimulatory effect on stem cells, further promoting tissue repair [103,104]. Based on this premise, Wang et al. proceeded to explore the encapsulation of pegylated EGF in PLGA nanoparticles as well as the formulation of hydrogels through the encapsulation of EPO within two-phase particles composed of PLGA and poly(sebacic acid). This system, through the epidermis to the ischemic lesion area, was observed to effectively traverse the BBB and stimulate the differentiation of endogenous neural stem cells. This ultimately facilitated the notable restoration of tissue and nerve function in the mice [105]. A prospective approach to treating ISs could involve preventing neutrophil infiltration. Song et al. developed a rod-shaped PLGA nanoparticle loaded with piceatannol for targeted intervention in neutrophil–endothelial cell interactions. They drew inspiration from the fact that neutrophils exhibit a preference for phagocytosing elongated particles and that rod-shaped PLGA is readily phagocytosed by neutrophils [106] (Figure 2a). This strategy can protect neurons and attenuate stroke damage to the brain by reducing neutrophil adhesion to endothelial cells and reducing immune infiltration (Figure 2b). In addition, PLGA can be used to form nanoparticles containing superparamagnetic iron oxide and Cy7.5, serving a pivotal role in MRI and fluorescence imaging [107].

Surface modification with PEG prolongs in vivo circulation and decreases the immunogenicity of nanoparticles [108]. Curcumin is known as an anti-inflammatory and antioxidant, but weak hydrophilicity and chemical instability make its application challenging [109]. Wang et al. used an amphiphilic copolymer consisting of PEG and PLA to prepare NDDS-encapsulating curcumin, which improves the stability of curcumin in blood circulation. The NDDS could protect the BBB by inhibiting the decrease in tight junction proteins [110]. The results showed that this NDDS inhibited M1 microglia polarization and inflammatory damage, thereby further promoting the functional recovery of brain tissue in mice. In another interesting study, PEG was used for modification to prepare pH-sensitive polymers loaded with rapamycin (RAPA), named RAPA@NPs. A pH-sensitive link was employed to achieve acid-triggered drug release, Ce6 was selected as the near-infrared imaging agent, and Gd^3+^ chelator was chosen for implementing bimodal imaging [111]. The results showed that the prepared RAPA@NPs not only overcame the drawbacks, such as the poor solubility of RAPA itself, thereby improving the therapeutic efficiency of RAPA, but also exhibited good biocompatibility and acid-enhanced bimodal imaging capabilities. RAPA@NPs preferentially aggregated at the ischemic site of the brain and achieved significant neuroprotective effects. The study provides a promising NDDS for drug tracking, treatment, and the early diagnosis of an IS as well as a reference towards the accurate imaging and treatment of other diseases [111]. Similarly, Ding et al. developed PEG coupled with urokinase (UK), wherein UK: PEG-UK was mixed in a 1:1 ratio [112]. The results demonstrated that this NDDS provided dual targeting of the macro-vasculature and the microcirculation, showing excellent neurologic function scores and smaller infarcts in the MCAO model area.

Dendritic macromolecules are excellent candidates for biological and pharmaceutical applications [113,114,115]. Among them, polyamidoamine dendrimer (PAMAM) is the most widely studied dendrimer [116]. This can be attributed to their controllable size structure, good water solubility, extensive internal drug-carrying space, and the potential for external modification [117,118]. However, PAMAM suffers from susceptibility to clearance by the reticuloendothelial system (RES) and a lack of targeting ability [119]. Despite these limitations, PAMAM can acquire enhanced capabilities through modifying and transforming into more desirable carriers. Examples include the conjugation of ligands such as PEG, folate analogs, protein analogs (transferrin, lactoferrin), amino acids, and peptides [120,121,122,123]. Successful NDDSs based on PAMAM for the treatment of neuroinflammation and ISs have been reported, yielding satisfactory therapeutic outcomes [124,125,126,127].

Collectively, the excellent properties of polymers allow them to be used as an NDDS, taking advantage of the microenvironmental characteristics for responsive release [111,128,129,130].

### 3.2. Inorganic Nanoparticles

Inorganic nanoparticles possess unique physicochemical properties [90]. They are characterized by a controllable structure, modifiable surface, and high loading efficiency, showing great research value in disease treatment and diagnosis [131,132,133]. For instance, gold nanoparticles are easy to synthesize and functionalize for versatile applications, possessing specific physical, electrical, magnetic, and optical properties [134,135]. Silica and iron have natural properties that allow them to work for MRI tracking [136,137]. Fe_3_O_4_ and CeO_2_ have enzyme-like activities that contribute positively to the removal of excess ROS [138,139,140].

Expanding on last paragraph, given the capability of gold nanoclusters to penetrate the BBB and the anti-inflammatory and antioxidant properties of dihydrolipoic acid, Xiao et al. synthesized functionalized gold nanoclusters carrying dihydrolipoic acid [141,142,143,144]. After the gold nanoclusters carried the drug through the BBB and into the brain, the dihydrolipoic acid underwent reduction and exerted its scavenging effects [144]. By regulating microglia and polarizing them to the M2 type, this NDDS can reduce the inflammatory response and improve the neuronal survival rate. In addition, the clinical application of gold nanoparticles for the photothermal therapy of prostate cancer has been carried out [145,146].

Magnetic iron oxide nanoparticles have garnered increasing attention due to their unique properties [94]. Liu et al. developed co-doped Fe_3_O_4_ nanoenzymes to ameliorate the RONS overload injury caused by an IS [147]. In vitro cellular experiments demonstrated that Fe_3_O_4_ nanoenzymes were effective in ameliorating neuroinflammation caused by ISs, as well as significantly reducing the infarct volume in both transient and permanent stroke models. Wang et al. developed an MRI visualization technique to track and visualize transplanted stem cells, offering valuable insights to enhance the effectiveness of IS treatment [148]. Similarly, some studies used superparamagnetic iron oxide nanoparticles (SPIO) to label stem cells for imaging tracking in IS therapy [107,149,150]. In addition, iron oxide nanoparticles can be loaded with dexamethasone and L-carnosine peptide for targeted delivery in ISs, which already yield satisfactory experimental results. It demonstrated the feasibility that iron oxide nanoparticles can be used as an NDDS for the treatment of ISs [151]. Clinical approval of iron oxide-based MRI contrast agents has been obtained in Europe and the United States; however, challenges persist in their clinical translation [152]. Overall, iron oxide-based nanoparticles hold excellent potential for application and clinical translation [153,154,155,156,157].

CeO_2_ nanoparticles are effective free radical scavengers [158,159]. Li et al. developed CeO_2_ nanoparticles loaded with butylphthalide (NBP-CeO_2_ NPs), which have neuroprotective effects for ISs [160]. The results of long-term experiments demonstrated that NBP-CeO_2_ NPs can promote vascular repair and improve the behavioral functions of mice. The combination of free radical scavenging and neurovascular repair can significantly reduce reperfusion injury, and this treatment approach holds promise for applications in ISs. He et al. developed CeO_2_ nanoparticles encapsulated with zeolitic imidazolate framework-8-capped (CeO_2_@ZIF-8). Compared to bare CeO_2_, the CeO_2_@ZIF-8 exhibited higher stability and longer blood circulation time under physiological conditions [161]. Moreover, due to its peroxidase-like properties, ZIF-8 enhances the free radical scavenging ability of CeO_2_@ZIF-8 compared to free CeO_2_, showing excellent preventive and therapeutic effects in neuroprotective therapy for ISs. Liao et al. introduced the concept of mitochondrial microenvironmental regulation and developed an NDDS based on CeO_2_-targeting mitochondria [162] (Figure 3). Mitigating oxidative stress injury can be achieved by regulating the mitochondrial microenvironment to promote ischemic recovery. 

In addition, manganese-based nanoparticles, including MnO_2_ and Mn_3_O_4_, have also received much attention in ISs [52,163,164]. In the news from 2023, during the European Congress of Radiology (ECR) in Vienna, Austria, the GE Healthcare Group announced that the completion of Phase I subject recruitment for its pioneering manganese-based macrocyclic magnetic resonance imaging (MRI) contrast agent in early clinical development program. To sum up, the integration of inorganic nanoparticles into ischemic treatment approaches holds great potential for enhancing therapeutic outcomes through the modulation of cellular processes and microenvironmental factors.

### 3.3. Liposomes

Liposomes, discovered by Bangham in 1965, are enclosed vesicles composed of ordered lipid bilayers [165]. The bilayers consist mainly of amphiphilic phospholipids with aqueous spaces inside. Thus, components with hydrophilic properties can be encapsulated in the aqueous cavity of the liposome, while components with hydrophobic properties can be encapsulated in the lipid bilayer [166] (Figure 4). Compared to other NDDSs, studies on liposomes are relatively well established. Due to their excellent biocompatibility, high safety profile, and noteworthy therapeutic efficacy, numerous liposome formulations have received FDA-approval, signifying the progression from laboratory investigation to clinical application [167,168]. Undoubtedly, the successful translation of liposomes from the laboratory to the clinic has garnered considerable attention from researchers across diverse fields, fostering further in-depth investigations into liposomes [169,170]. 

Plain liposomes are easily cleared by the reticuloendothelial system (RES), while PEGylated liposomes extend the drug’s circulation time in the bloodstream and mitigate the risk of recognition and clearance by immune cells [171]. Thomas et al. prepared a PEGylated liposome loaded with atorvastatin, which could accumulated at the ischemic lesion area effectively, reducing infarct volume and promoting neurological recovery [172]. The use of PEGylated liposomes for delivering neuroprotective agents also improves drug bioavailability in vivo. Notable examples include citicoline [173], lycopene [174], FK506 [175], and plasminogen activators [176]. In addition, the addition of some FC fragments, transferrin, and other kinds of ligands or stimuli-responsive fragments to liposomes can achieve active targeting and responsive release [177,178,179]. To address the phenomenon of neutrophil infiltration into the ischemic zone during an IS, resulting in the release of neutrophil extracellular traps (NETs) and subsequent neuronal damage, Sun et al. developed a smart liposome with ischemic lesion targeting and ROS-responsive release [180]. The results showed that the smart liposome could regulate NETs and promote microglia transformation to the M2 type to treat ISs, resulting in a cerebral infarction area approximately one-fourth that of the saline group. Li et al. investigated methods to enhance the delivery efficiency of the commercially available drug ginkgolide B (GB) by developing a liposomal formulation that binds GB to high-lipophilic docosahexaenoic acid (DHA) (Lipo@GB-DHA) [181]. The results indicated that when compared with the free GB group, the amount of GB in the target ischemic hemisphere in the Lipo@GB-DHA group was 2.2 times higher than that in the free GB group. Additionally, the Lipo@GB-DHA group showed a smaller infarct area, better inhibition of neuronal apoptosis, and improved recovery of neurological function. Yao et al. developed a pH-responsive fluorescent liposome probe and established a correlation between fluorescent imaging and neurologic deficiency scores. The approach provided a novel approach for assessing the extent of ISs in different acidic microenvironments [182]. Noteworthly, the transnasal administration of liposomes have achieved good therapeutic results in rat IS models [183,184].

In conclusion, substantial advancements have been achieved in utilizing liposomes as an NDDS. However, problems such as instability, leakage or poor targeting, and the sudden release of drugs remain to be solved in liposomes [185]. These issues highlight the critical need for the development of advanced technologies aimed at enhancing the properties of liposomes.

### 3.4. Cell Membrane-Coated Nanoparticles

Targeted nanomedicines hold promising potential in stroke therapy. However, some of them are intrinsically foreign and face the risk of being removed by the RES [186]. In order to address this obstacle, the researchers proposed a cell membrane-coated strategy. Cell membrane-coated nanoparticles possess the unique advantage of imparting biological characteristics by hiding traditional nanoparticles in natural cell membranes [187]. By selecting different kinds of cell membranes to modify the outer layers of NDDSs, NDDSs with enhanced surface functionality can be created to achieve diverse goals [188] (Figure 5). Cell membranes from different sources have different functional characteristics, as shown in Table 1. However, cell membranes still have some limitations [189]. Fortunately, it is possible to modify some targeting proteins or peptides on the membrane surface to further improve the targeting performance and ultimately achieve better therapeutic effects [190,191].

#### 3.4.1. RBC Membrane-Coated Nanoparticles (RBC-NPs)

It is well known that RBCs are abundantly present within the blood and lack nucleus and mitochondria [189]. Consequently, the RBC membrane is easily extracted. The lifespan of RBCs in the body is about four months. During this time, the surface expression of the CD47 protein, which serves as a “do-not-eat-me” signal, prevents RBCs from being removed by the RES [203]. These characteristics of RBCs offer potential prerequisites for the development of RBC-NPs [204].

RBC-NPs were first reported by Zhang et al. in 2011 [205]. Compared with NPs coated with hydrophilic polymer PEG, which showed long circulation in vivo, the RBC-NPs retained RBCs’ biological properties, showing a longer circulation time. After that, researchers proposed diverse approaches to modify targeting peptides or ligands on the RBC membrane, which could not disrupt the RBC membrane but improve the brain targeting ability [206,207,208].

Shi et al. developed an engineered RBC-NP named Mn_3_O_4_@nanoerythrocyte-T7 (MNET) with smart oxygen regulation and free radical scavenging [209] (Figure 6a). Hemoglobin (HB) in RBCs underwent oxygen uptake and release, functioning like an oxygen sponge. Meanwhile, Mn_3_O_4_ NPs showed high biocompatibility and multiple antioxidant enzyme activities. By combining the advantages of RBCs, Mn_3_O_4_, and T7 peptides, the rescue of the ischemic microenvironment was finally realized using MNET. More excitingly, MNET could be used for continuous treatment via the oxygen spongy effect of HB. During an IS, the oxygen spongy function of HB can provide oxygen to the hypoxic area and reduce ischemic injury (Figure 6b). Oxygen reperfusion after thrombolysis causes the production of free radicals, and HB could play an important role in absorbing excess oxygen and reducing oxidative stress. Liu et al. also developed an integrated approach for acute ISs inspired by the oxygen spongy properties of HB [210]. They corrected abnormalities in glucose metabolism and provided energy to neurons by releasing methoxatin, which acts to activate the cellular Akt/GSK-3β pathway. It can be observed that the oxygen balance of the microenvironment and glucose metabolism are important for neuronal recovery. Lv et al. designed an erythrocyte membrane delivery system (SHp-RBC-NP/NR2B9C) with cerebral ischemic region-targeting and ROS-responsive release capabilities [211]. Based on the fact that ROS are released in large quantities from the ischemic region during the pathogenesis of an IS, phenylboronic acid with ROS-responsive release capabilities was added to the SHp-RBC-NP/NR2B9C. It was verified in the rat MCAO model that the SHp-RBC-NP/NR2B9C could successfully reach the ischemic lesion, release the neuroprotectant NR2B9C in the ischemic ROS microenvironment responsively, and reduce the volume of cerebral infarcts.

In conclusion, RBC-NPs, as the earliest membrane-coated NDDS studied, have tremendous possibilities through modification [212]. They have aroused researchers’ interest, and more researchers have begun to exploit the feasibility of fabricating hybrid RBC-NPs.

#### 3.4.2. Platelet Membrane-Coated Nanoparticles

Platelets have a shorter lifespan compared to erythrocytes and are present in the blood in smaller numbers than erythrocytes [213]. Platelets, in addition to CD47, exhibit other expression clusters, such as CD55 and CD59, on their membrane surface, preventing phagocytosis and clearance by immune cells, consequently enhancing immune evasion [214]. In addition, platelets are known for their role in recognizing and repairing vascular damage and responding to inflammation. Consequently, nanoparticles encapsulated in platelet membranes can be directed toward the site of the injury for targeted delivery [215].

A precedent for biomimetic platelet membrane-coated nanoparticles, which mimic platelets and evade the immune system, was established by Zhang et al. in 2015 [193]. Since then, there has been a growing focus on platelet membrane-coated nanoparticles. For example, Cui et al. selected GB, a neuroprotective agent with anti-inflammatory and antioxidant properties, and coated GB with platelet membranes (PM-GB) [216]. Because platelet membranes could target inflammatory injuries, the concentration of GB at the injury site increased, resulting in improved drug delivery to the lesion. The results demonstrated that PM-GB was suitable for the treatment of ISs by inhibiting oxidative stress and reducing iron-related cell death. Zhao et al. designed a platelet membrane-coated nanoparticle with multi-function for the therapy of ISs [217] (Figure 7a). To be specific, when a platelet membrane adheres to inflamed neutrophils, platelet membrane-coated nanoparticles can then follow the neutrophils into the inflamed area. The platelet membrane-coated nanoparticles consisted of T7 peptide, PHis (an acid-responsive fragment), and MiRNA-Let-7c drugs. Based on the multiple functional fragments, this NDDS could be BBB targeting and swell to release miRNA to inhibit M1 cell polarization. Li et al. developed platelet-derived bio-nanobubbles with integrated diagnostic capabilities using platelet vesicles [218]. In addition to platelet vesicles, the formulation also includes γ-Fe_2_O_3_ and the NO precursor (L-arginine). The results showed that it could increase the flow of blood to the lesion site, prolong the treatment window through the vasodilatory effect of NO, and provide auxiliary diagnostic imaging [219] (Figure 7b). After that, Li further conducted the therapeutic mechanism of PAMNs. The results illustrated that it can rapidly dilate vessels and improve vascular flow, which is beneficial for the early-stage therapy of ISs (Figure 7c).

To sum up, the properties of platelets allow for platelet membrane-coated nanoparticles to be widely used for vascular embolism disease [220,221,222,223,224,225,226]. 

#### 3.4.3. WBC Membrane-Coated Nanoparticles

WBCs are important blood cells of the body’s immune system, encompassing various types, such as NK cells, neutrophils, macrophages, and lymphocytes. As the body’s guardians, WBCs protect the body from disease and arrive at the injury site immediately when an IS occurs [227]. In contrast to RBCs and platelets, WBCs have nuclear structures and are less abundant, rendering their membrane isolation relatively challenging. To be noted, natural WBC membranes without modification have the capability of targeting inflammatory sites and tumor tissue [228,229]. 

Here, we mainly discuss membranes from neutrophils and macrophages to fabricate NDDSs. Neutrophils are virtually absent from the brain, but after an IS occurs, neutrophils rapidly increase in numbers and enter the ischemic lesion area in a short time [230]. The utilization of magnetic probes coated with neutrophil membranes holds promise for imaging neuroinflammation in ISs [231]. Neutrophil membranes or extracellular vesicles can be used to intracerebrally target, both of which have satisfactory effects in ISs [232,233]. Feng et al. treated ISs with neutrophil membrane-encapsulated Prussian blue nanoenzyme (MPBzyme@NCM) [234] (Figure 8a). The results showed that MPBzyme@NCM could polarize microglia from the M1 phenotype to the M2 phenotype, reduce inflammatory responses, and protect injured brain tissue (Figure 8b). This strategy showed promise in extending its potential application to other CNS illnesses.

Macrophages are immune cells with the capabilities of pathogen recognition and phagocytic clearance. At the onset of an inflammatory response, macrophages exhibit a tendency towards inflammation and migrate to the inflammatory site to phagocytose and eliminate pathogens [235,236]. Based on these remarkable features, macrophage membrane-coated nanoparticles were widely applied in delivery studies [237]. Li et al. developed mesoporous SiO_2_ nanoparticles loaded with the neuroprotectant FTY-720 (MnO_2_ + FTY) and then wrapped MnO_2_ + FTY with macrophage membrane vesicles to treat ISs [52] (Figure 9a). In this system, macrophage membranes confer nanoparticles with the ability to target inflammatory lesions. MnO_2_ nanoparticles have broad surface and CAT properties, which can effectively scavenge excess ROS and promote O_2_ conversion, thereby reducing inflammatory responses and rescuing dying neurons. FTY-720 reverses the pro-inflammatory microenvironment (Figure 9b). Su et al. used macrophage membranes to encapsulate curcumin for treating ISs, yielding promising therapeutic outcomes. This study offers valuable insights in combining the traditional Chinese medicine and modern technology [238].

In conclusion, the utilization of WBC membrane-coated nanoparticles emerges as a highly promising avenue for the treatment of both inflammation and cancer. Despite the significant advancements made in this field, further exploration is warranted to fully harness the capabilities of WBC membrane-coated nanoparticles.

#### 3.4.4. Cancer Cell Membrane-Coated Nanoparticles

Cancer cells exhibit immune evasion and homology-targeting capabilities; thus, nanoparticles encapsulated within cancer cell membranes can be used for targeted therapy [197,239,240]. This strategy has been widely used in cancer homologous-targeted therapy and cancer vaccine development [241,242,243,244,245]. Apart from cancer, He et al. found that cancer cell membrane-coated nanoparticles could be applied to IS treatment. As shown in Figure 10, they developed a novel biomimetic nanoplatform, termed MPP/SCB, by cloaking a succinobucol-loaded pH-sensitive polymeric nanovehicle with a 4T1 cell membrane. They drew inspiration from the BBB-penetrating ability of 4T1 cancer cells during brain metastasis. The primary factors contributing to this phenomenon include the heightened affinity of certain adhesion molecules highly expressed on the membrane of 4T1 cells, enabling adhesion to leukocytes, endothelial cells, and platelets [246]. MPP/SCB significantly improved microvascular reperfusion in the ischemic hemisphere, leading to a remarkable 69.9% reduction in infarct volume and demonstrating superior neuroprotective effects compared to uncamouflaged PP/SCB. Although MPP/SCB show negligible biotoxicity, the potential presence of numerous tumor antigens on cancer cell membranes poses an unknown risk that warrants further investigation and validation. Their findings highlight the potential of cancer cell membrane-coated nanoparticles for the targeted therapy of cerebral ischemic lesions in ISs and inspire us to explore additional functions of cancer cell membranes beyond their interaction with cancer cell homologous targets.

#### 3.4.5. Other Cell Membrane-Coated Nanoparticles

Stem cells, characterized by their capacity for self-renewal and differentiation, exhibit the proficient recognition and repair of damaged tissues, homing ability, inflammation suppression, and tumor-targeting abilities [247]. Stem cell studies in treating ISs are extensive, with a predominant focus on the transplantation of stem cells or the administration of stem cell-derived vesicles and trophic cytokines [248,249,250,251,252,253]. At present, fewer studies have been conducted on stem cell membrane-coated nanoparticles for IS therapy [254]. We notice that there was a study using stem cell membrane-coated nanoparticles as a delivery platform, which leveraged the SDF-1/CXCR4 pathway to enhance targeted drug delivery in ISs [255]. By encapsulating glyburide-loaded PLGA within stem cell membranes, Ma et al. significantly improved stroke treatment efficacy [256]. This innovative approach not only underscores the importance of the SDF-1/CXCR4 axis in cell migration and homing but also offers a promising strategy for enhancing intracerebral drug delivery. 

Bacterial membrane-coated nanoparticles have been broadly used for targeted tumor therapy, antimicrobial therapy, vaccine development, and so on [257,258,259,260,261]. Although it has been reported that anaerobic bacteria can cross the BBB to treat gliomas [262], we did not find any reports of treating ISs with nanoparticles coated with bacterial membranes. 

## 4. Summary and Perspective

ISs are the second leading cause of death in the world, which is attributed to the narrow therapeutic window and the complexity of the disease progression involving multiple mechanisms, including neuro-excitotoxicity, oxidative stress, neuroinflammation, mitochondrial damage, and so on. Understanding how an IS occurs is a necessary prerequisite for attempting to resolve the disease. Thrombolysis is the preferred treatment to prevent neuronal damage. The prognosis after thrombolysis is key to restoring normal limb function. 

With the rapid development of nanotechnology, polymer nanoparticles, inorganic nanoparticles, and liposomes, membrane-coated nanoparticles have gradually emerged in diagnosing and treating numerous diseases, including ISs [263,264,265,266]. Some have successfully transitioned from the laboratory to clinical applications. However, there are still some problems that limit their application. In general, polymers increase drug stability and can be surface modified to enhance targeting or improve biocompatibility. However, the relatively high cost and complex preparation process limit their mass production. Some inorganic nanoparticles exhibit exceptional photothermal and imaging properties, but as exogenous materials, they still encounter challenges related to unknown cytotoxicity and complex degradation issues. Among them, magnetic nanoparticles show promising prospects, and functionalized magnetic nanoparticles are the trend of future development. Additionally, optimizing their multifunctional imaging capabilities holds significant potential for advancing the clinical diagnosis and treatment of diseases [267,268]. Liposomes have notably progressed compared to other nanoparticles, boasting advantages such as low toxicity and high biocompatibility. Various drugs have been successfully delivered using liposomes. However, liposomes are expensive, which can exacerbate the financial burden on patients. Membrane-coated nanoparticles have emerged as a novel approach for targeted drug delivery in recent years. It is characterized by the use of biomimetic membranes, where the drug is camouflaged as an endogenous substance. This strategy helps evade the RES clearance, thereby extending the drug’s half-life. Moreover, membrane-coated nanoparticles tend to exhibit targeting capabilities, enhancing the efficacy of precise therapy. The experimental results have yielded promising outcomes, suggesting a bright future ahead. However, these findings are still in their infancy, necessitating further investments. The translation from the laboratory to clinical settings necessitates the careful consideration of immune rejection, particularly regarding proteins or genes carried by biomembranes sourced from different origins. In addition, the fusion of multiple membranes is emerging as a trend. Challenges such as difficulty in scale-up production, uncertainty in variant proteins in membranes, and ensuring the controllable preparations necessitate ongoing research. Nevertheless, it is undeniable that membrane-coated nanoparticles have inherent advantages and hold broad potential for application.

Overall, an IS is a complex pathophysiological process, in which the interference of the BBB and other factors significantly impacts drug delivery. NDDSs struggle to deliver drugs effectively within the brain. Most studies involving NDDSs are still in the laboratory stage, and the translation of NDDSs into clinical practice remains a daunting challenge. 

## Figures and Tables

**Figure 1 molecules-29-01848-f001:**
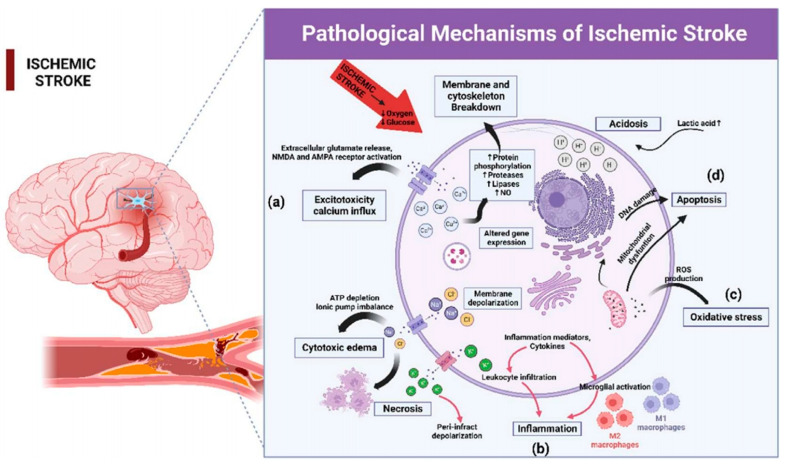
Pathologic mechanisms and cascading injury processes in ISs [35]. (**a**) Calcium overload leads to neuroexcitotoxicity; (**b**) Inflammation; (**c**) Mitochondrial dysfunction leads to oxidative stress; (**d**) Apoptosis; Copyright 2023, Elsevier.

**Figure 2 molecules-29-01848-f002:**
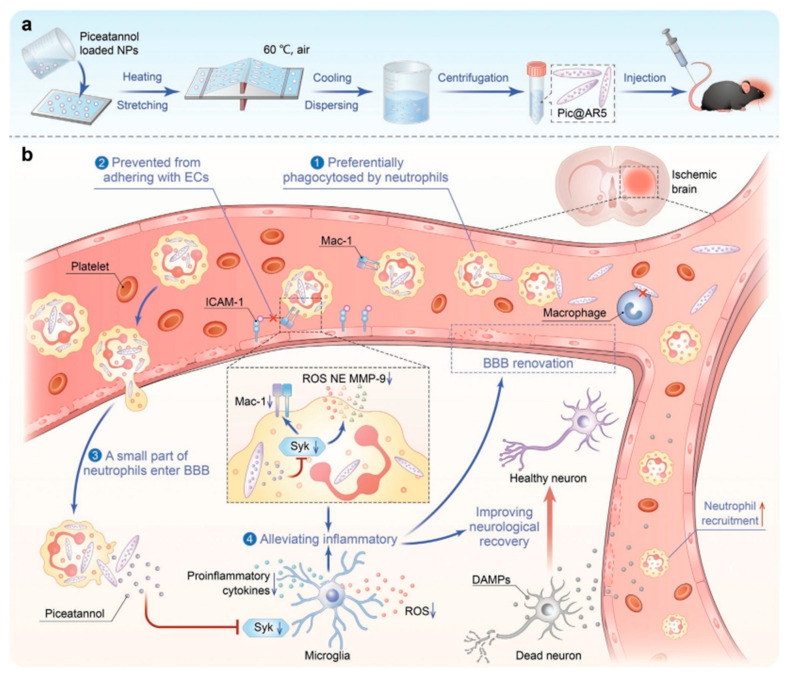
(**a**) Preparation of rod-like nanoparticles by solvent evaporation and film stretching methods; (**b**) Rod-like nanoparticles (Pic@AR5) were specifically phagocytosed by neutrophils, followed by the release of pisaclitaxel which blocked the Syk pathway and reduced the expression of β2 integrins, thus preventing the adhesion of most neutrophils to endothelial cells. Meanwhile, a small number of Pic@AR5-carried neutrophils enter the BBB and release piceatannol in the ischemic zone, inhibiting the Syk pathway in microglia and attenuating microglia-mediated neuroinflammation [106]. Copyright 2023, Wiley-VCH.

**Figure 3 molecules-29-01848-f003:**
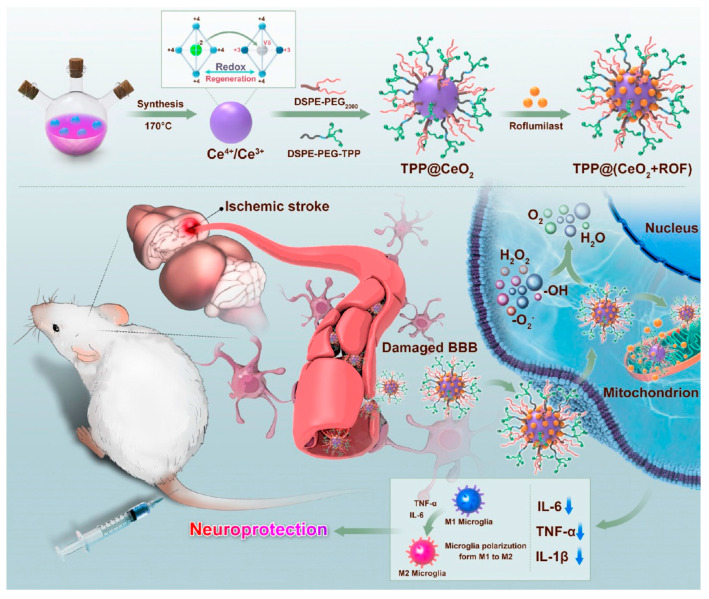
Schematic illustration of mitochondria-targeted cerium nanoenzymes for the therapy of ISs. Neuroprotective effects are achieved by attenuating oxidative stress and modulating microglia phenotype [162]. Copyright 2024, American Chemical Society.

**Figure 4 molecules-29-01848-f004:**
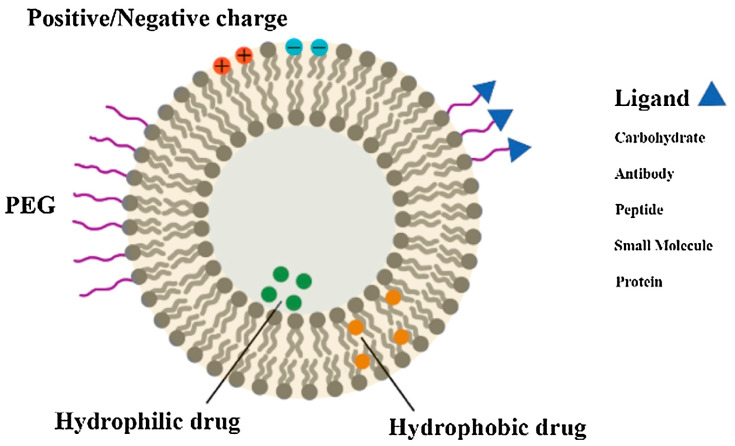
Schematic illustration of the functionalization of liposomes [166]. Copyright 2019, Elsevier.

**Figure 5 molecules-29-01848-f005:**
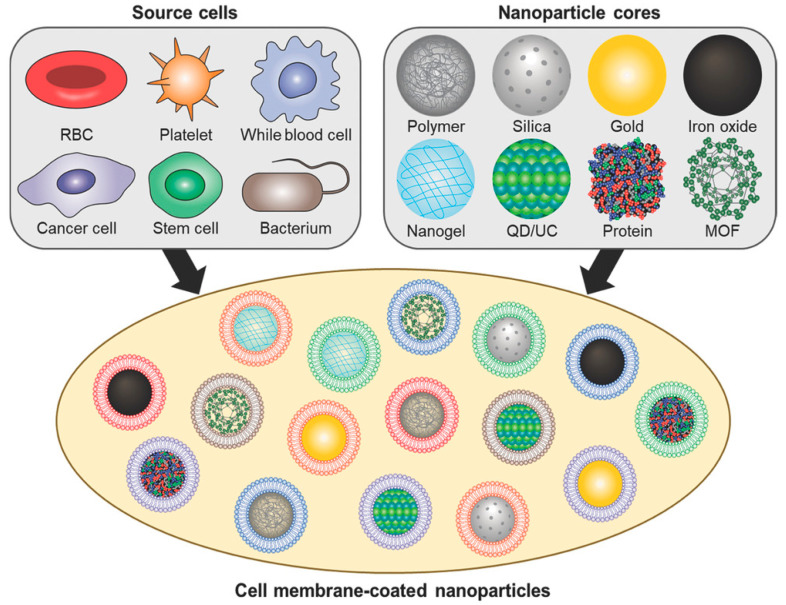
Diverse cell membrane-coated nanoparticles [188]. Copyright 2018, WILEY-VCH.

**Figure 6 molecules-29-01848-f006:**
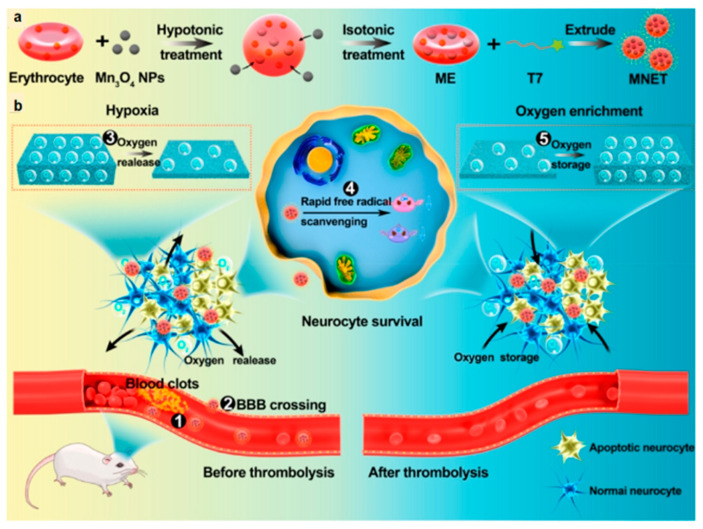
(**a**) Flowchart of MNET preparation; (**b**) Schematic illustration of oxygen sponge function and free radical scavenging of MNET in vivo. MNET crosses the BBB and accumulates at the site of infarction, with effects of releasing oxygen to relieve hypoxia and storing oxygen in a pro-oxidant environment after thrombolysis [209]. Copyright 2020, American Chemical Society.

**Figure 7 molecules-29-01848-f007:**
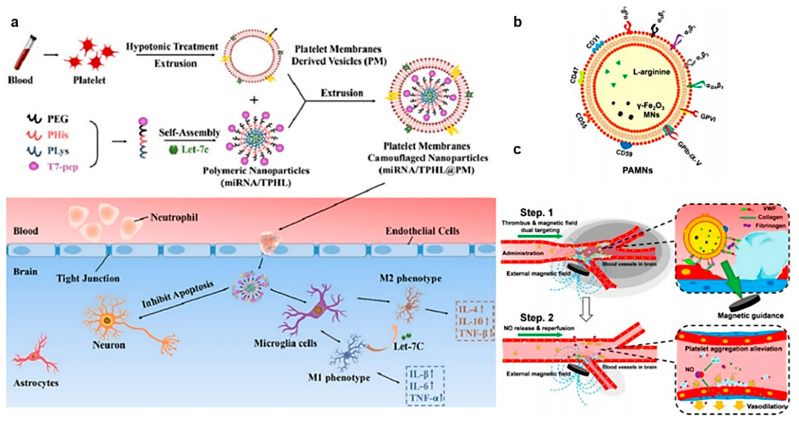
(**a**) Flowchart of miRNA/TPHI@PM preparation and how it functions. Delivery of miRNA through platelet membrane-coated nanoparticles to sites of brain lesions aims to reduce neuronal apoptosis and regulate microglial cell phenotype [217]. Copyright 2022, Elsevier Ltd. (**b**) The structure of PAMNs [219]. Copyright 2021, American Chemical Society. (**c**) Schematic illustration of the therapeutic mechanism of PAMNs. Due to the endowed platelet membrane function and external magnetic targeting, PAMNs rapidly reaches the ischemic region of stroke and achieves rapid targeted delivery of l-arginine. Meanwhile, NO generation in situ allows the induction of vasodilation and the reduction of PLT aggregation [219]. Copyright 2021, American Chemical Society.

**Figure 8 molecules-29-01848-f008:**
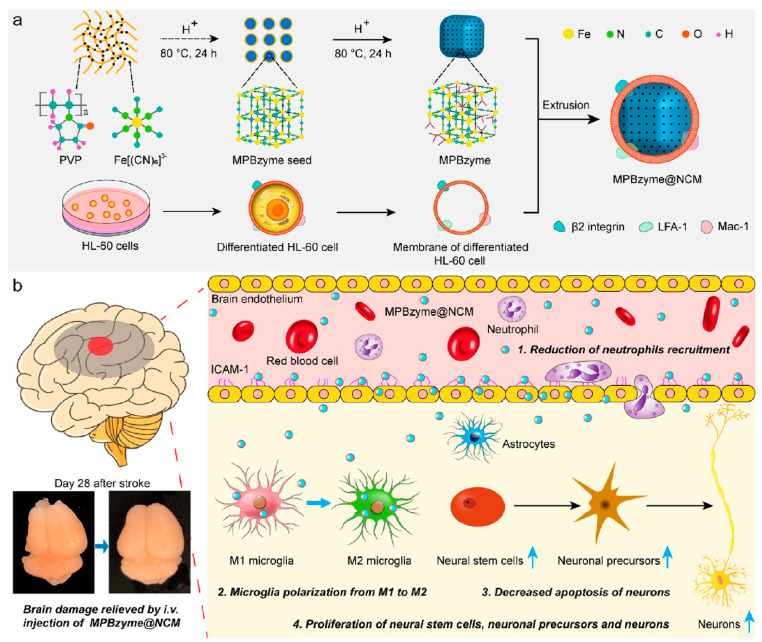
(**a**) Preparation of MPBzyme@NCM; (**b**) The effect of MPBzyme@NCM in vivo. Long-term therapeutic effects of stroke were exerted through the ability to reduce neutrophil recruitment, regulate microglia polarization from M1 to M2, reduce neuronal apoptosis, and increase neural stem cell proliferation [234]. Copyright 2021, American Chemical Society.

**Figure 9 molecules-29-01848-f009:**
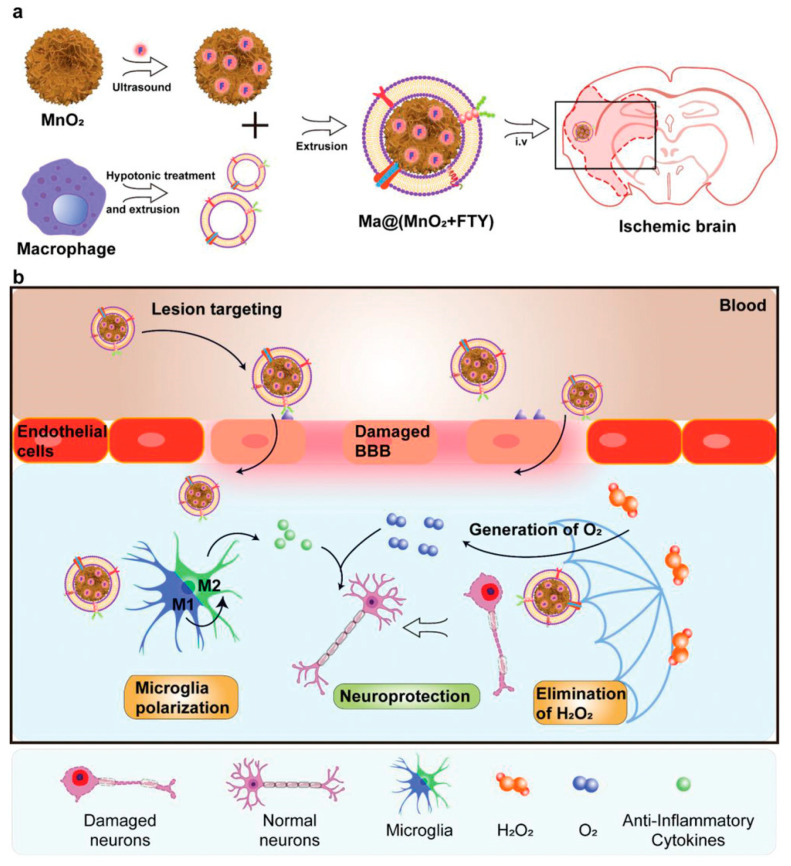
(**a**) Preparation of Ma@(MnO_2_ + FTY); (**b**) Schematic illustration of the effect of Ma@(MnO_2_ + FTY) [52]. Copyright 2021, Wiley-VCH.

**Figure 10 molecules-29-01848-f010:**
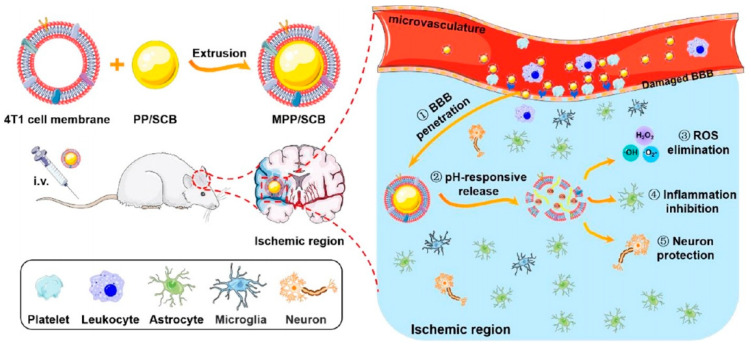
Preparation of MPP/SCB and its effect on ISs [246]. Copyright 2021, American Chemical Society.

**Table 1 molecules-29-01848-t001:** A summary of the key features of various cell membranes.

Cell Membrane Source	Key Features	Function	Reference
Red blood cell (RBC)	Immune evasion Easy to extract Surface expression of CD47	Prolonged circulation	[192]
Platelet	Specific targeting of damaged tissue Adherence to inflammatory neutrophil Surface expression of CD47, CD55 and CD59	Injury sites targeting	[193]
White blood cell (WBC)	Specific targeting of inflammatory tissue Endothelial adherence Adhesion at tumor sites Penetration of the BBB	Tumor and inflammatory site targeting	[194,195,196]
Cancer cell	Immune evasion Homologous targeting Anti-tumor ability	Tumor targeting	[197]
Stem cell	Immune evasion Tumor-specific properties Homing ability	Tumor targeting Inflammatory damage targeting	[198,199]
Bacteria	Promoting adaptive immunity	Tumor anaerobic targeting	[200,201,202]

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
