# Peer review of "Advanced Nano-Drug Delivery Systems in the Treatment of Ischemic Stroke"

_molecules, 2024, doi:10.3390/molecules29081848_

Round 1

Reviewer 1 Report

Comments and Suggestions for Authors

The authors have provided a thorough depiction of the present state of nanotherapeutics for SI. Nevertheless, before considering the manuscript for publication, certain aspects need to be attended to.

The subsection on inorganic nanoparticles and polymer nanoparticles requires a more detailed description, particularly regarding the clinical effects, which are only marginally addressed. This aspect should be elaborated upon further.

Author Response

Thanks for the comment. In the latest manuscript, we have added the discussion on polyethylene glycol (PEG) and polyamidoamine dendrimer (PAMAM) to the subsections of inorganic nanoparticles and polymer nanoparticles. For the clinical effects, we have only identified clinical applications of inorganic nanoparticles for assisted diagnostic imaging for IS. The clinical application of polymeric nanoparticles appears to be yet to be discovered. If the reviewer is aware of any related clinical applications, we welcome the input. Relevant revisions have been marked using the “Track Changes” function of MS word. 

Reviewer 2 Report

Comments and Suggestions for Authors

Peer Review of "Advanced nano-drug delivery systems in the treatment of ischemic stroke" by Jiajie Zhang, Zhong Chen, and Qi Chen

1. Brief Summary:

The paper provides a comprehensive overview of the challenges in treating ischemic stroke and the potential of nano-drug delivery systems as a solution. It highlights the importance of addressing reperfusion injury in ischemic stroke and discusses the advantages of functionalized nano-drug delivery modalities. The strength of the paper lies in its clear presentation of the research progress in this field.

2. General Concept Comments:

2a. Article:

The paper effectively outlines the potential of nano-drug delivery systems in treating ischemic stroke. However, it would benefit from a more detailed discussion on the limitations and challenges associated with these systems. Additionally, further clarification on the methodology used in nano-drug delivery studies would enhance the scientific rigor of the paper.

2b. Review:

The review provides a thorough coverage of the topic of nano-drug delivery systems in ischemic stroke. It effectively identifies the gap in knowledge regarding the inefficiency of conventional treatments and the need for targeted drug delivery. The references cited are appropriate and support the arguments presented in the paper. However, more recent references could be included to strengthen the relevance of the review.

3. Special Comments:

The paper successfully highlights the potential of nano-drug delivery systems in improving the treatment of ischemic stroke. To enhance the impact of the research, the authors could consider discussing the translational potential of these systems and any ongoing clinical trials in this area. Additionally, addressing the scalability and cost-effectiveness of nano-drug delivery systems would provide valuable insights for future research and development efforts.
